# RRM: Robust Reward Model Training Mitigates Reward Hacking

**Tianqi Liu**[1],[*] **Wei Xiong**[2],[†] **Jie Ren**[1], **Lichang Chen**[3],[†] **Junru Wu**[1], **Rishabh Joshi**[1], **Yang Gao**[1], **Jiaming Shen**[1], **Zhen Qin**[1], **Tianhe Yu**[1], **Daniel Sohn**[1], **Anastasiia Makarova**[1], **Jeremiah Liu**[1], **Yuan Liu**[1], **Bilal Piot**[1], **Abe Ittycheriah**[1], **Aviral Kumar**[1], **Mohammad Saleh**[1]
Google DeepMind[1], University of Illinois Urbana-Champaign[2], University of Maryland, College Park[3]

## ABSTRACT

Reward models (RMs) play a pivotal role in aligning large language models (LLMs) with human preferences. However, traditional RM training, which relies on response pairs tied to specific prompts, struggles to disentangle prompt-driven preferences from prompt-independent artifacts, such as response length and format. In this work, we expose a fundamental limitation of current RM training methods, where RMs fail to effectively distinguish between contextual signals and irrelevant artifacts when determining preferences. To address this, we introduce a causal framework that learns preferences independent of these artifacts and propose a novel data augmentation technique designed to eliminate them. Extensive experiments show that our approach successfully filters out undesirable artifacts, yielding a more robust reward model (RRM). Our RRM improves the performance of a pairwise reward model trained on Gemma-2-9b-it, on Reward-Bench, increasing accuracy from 80.61% to 84.15%. Additionally, we train two DPO policies using both the RM and RRM, demonstrating that the RRM significantly enhances DPO-aligned policies, improving MT-Bench scores from 7.27 to 8.31 and length-controlled win-rates in AlpacaEval-2 from 33.46% to 52.49%.

## 1 INTRODUCTION

Reinforcement Learning from Human Feedback (RLHF) has become a cornerstone in aligning large language models (LLMs) with human preferences to produce responses that are more helpful, honest, and harmless (Ouyang et al., 2022; Bai et al., 2022a). This approach involves training a reward model (RM) on human feedback, which then guides the LLM to generate high-quality responses through reinforcement learning. The success of RLHF is evident in various AI systems, such as Gemini (Team et al., 2023) and GPT-4 (Achiam et al., 2023). Despite its effectiveness, RLHF faces the fundamental issue of reward hacking (Gao et al., 2023), where the model maximizes the reward function without truly aligning with the intended human preferences. This hacking issue occurs because the RM, while a powerful tool, is an imperfect proxy for human judgment and often struggles with out-of-distribution generalization (Eisenstein et al., 2023).

The reward hacking problem manifests in several ways, with verbosity being a common issue: LLMs tend to generate longer responses to appear more detailed or explanatory, exploiting human raters' bias towards lengthier content (Shen et al., 2023b; Singhal et al., 2023). In recognition of this challenge, extensive efforts have been made in the literature. ODIN (Chen et al.) designs a two-head approach to learn the quality reward that is orthogonal to length. Similarly, length-controlled Alpaca (Dubois et al., 2024a) estimates the controlled direct effect (VanderWeele, 2011) through logistic regression by adjusting the length. To mitigate the length bias, an improved version (Park et al., 2024) of DPO (Rafailov et al., 2024) introduces length as penalty to the reward score. In practice, there are more reward hacking patterns beyond length, such as format (markdowns, bold-faces) and patterns (certain $n$-grams or emojis). This is largely due to the large output space of language with limited preference data, as well as the diverse and subjective nature of human preferences.

---

[*]Correspondence to Tianqi Liu, `tianqiliu@google.com`.
[†]Work done during an internship at Google DeepMind.

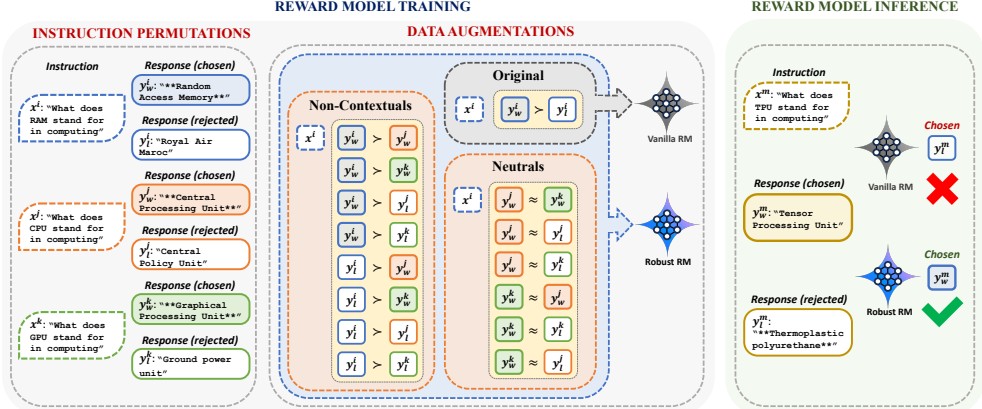

Figure 1: The pipeline of our proposed robust reward model (RRM), which aims to decouple contextual preference quality signal and context-free artifacts. Suppose a proportion of chosen responses have certain artifact (bold-face wrapped with "∗∗" in this figure), the reward model can hack the pattern and choose the response with the artifact instead of carefully reading the prompt. With our data augmentations, we can effectively balance the context-free artifacts in chosen and rejected responses, thus ensuring a more robust reward model during inference.

It is challenging to identify and mitigate all potential exploitation patterns. We may consider the causal perspective to explain this phenomena. Given a prompt $x$ and a pair of responses $(y_1, y_2)$, the human preference can be caused by the real quality $s(x, y_1, y_2)$ that is associated with the prompt, or by the context-free artifacts $a(y_1, y_2)$ in the responses that do not depend on prompt. Traditional reward model training cannot differentiate the above two factors. There are two reasons for this. First, the pair of responses are always contextual and on-topic to the prompt, thus no counterfactual prompt (prompt from another examples) is used. The reward model may learn the artifacts existing in the responses by ignoring the prompt. If we use the counterfactual prompt, it can help estimate the level of artifact bias ($\mathbb{P}(y_1 \succ y_2 | x')$ with $x' \neq x$) existing in the preference dataset (Zhao et al., 2021). Second, even if we adjust a few common artifacts, not all artifacts are observable and thus there is no easy way to control all the artifacts explicitly to answer the question "what will the preference be if both responses share the same artifacts?".

In response to these challenges, we propose a simple and effective method to improve reward modeling. We first formulate the reward model training in a causal framework, then we augment the reward model training data based on the causal rules. By doing so, we can effectively adjust the artifacts and only learn the real quality. Our pipeline is illustrated in Figure 1, where we augment the reward model training data by using responses from other examples to effectively balance the artifacts in chosen and rejected responses. To summarize, the contributions of this paper are three-fold:

- We identify a key issue with traditional reward model training: it often fails to distinguish between genuine contextual preference signals and spurious context-free artifacts.
- To address this, we propose a causal graph for human preference modeling and introduce data augmentation to mitigate artifacts learned by the reward model.
- We further demonstrate that policies trained on these robust reward models consistently outperform those based on baseline reward models.

## 2 PRELIMINARIES

In preference learning, we assume that there exists a preference oracle that determines the probability $\mathbb{P}(y_1 \succ y_2 | x)$ that response $y_1$ is preferred over $y_2$ given the prompt $x$. Our goal is to optimize the preference by querying the preference oracle within certain budget constraint. In what follows, we first review the major ways to approximate and estimate the oracle based on a human preference dataset $\mathcal{D}_{hf} = \{x^{(i)}, y_w^{(i)}, y_l^{(i)}\}_{i=1}^{N}$. where $x^{(i)}$ represents prompt for example $i$, and $(y_w^{(i)}, y_l^{(i)})$ represents the chosen and rejected response, respectively.

**Reward models** Bradley-Terry pointwise reward model (Bradley & Terry, 1952; Ouyang et al., 2022) is a widely adopted method, which additionally assumes that there exists a reward function $r(x, y) \in \mathbb{R}$ and the preference oracle satisfies

$$\mathbb{P}(y_1 \succ y_2 | x) = \frac{\exp(r(x, y_1))}{\exp(r(x, y_1)) + \exp(r(x, y_2))} = \sigma\big(r(x, y_1) - r(x, y_2)\big).$$

Then, we can fit the Bradley-Terry model by maximizing the log-likelihood on the training set:

$$\mathcal{L}(r_\phi, \mathcal{D}_{\text{hf}}) = -\mathbb{E}_{(x, y_w, y_l) \sim \mathcal{D}_{\text{hf}}} \left[ \log \sigma \left( r_\phi(x, y_w) - r_\phi(x, y_l) \right) \right]. \tag{1}$$

The second predominant approach is the pairwise ranking model (Zhao et al., 2023; Jiang et al., 2023), which takes a prompt and a pair of responses as the input, and directly predicts the probability $\mathbb{P}(y_1 \succ y_2 | x)$, which subsumes the BT model as a subclass. In the literature, the pairwise preference model has shown to outperform pointwise BT reward both empirically (Zhao et al., 2023; Jiang et al., 2023; Dong et al., 2024) and theoretically (Ye et al., 2024) due to its flexibility and larger function class capacity. Specifically, we denote the pairwise ranking model and leverage the next token prediction ability of the language model to format the sample as:

"[CONTEXT] $\{x\}$ [RESPONSE A] $\{y_1\}$ [RESPONSE B] $\{y_2\}$"

Then, the model outputs either "A" or "B" as preferred one. We use the probability of decoding "A" as estimation of the preference probability $\hat{\mathbb{P}}(y_1 \succ y_2 | x)$[1]. In this work, we use the pairwise ranking model for its superior performance and flexibility.

**Alignment Algorithms** Start with a reward function $r(x, y)$, a reference policy $\pi_{\text{ref}}$, and input prompt distribution $\mathcal{P}$, a policy $\pi$ is trained to optimize for the following objective:

$$\max_\pi \mathbb{E}_{x \sim \mathcal{P}} \left[ \mathbb{E}_{y \sim \pi(\cdot|x)} r(x, y) - \beta \mathbb{D}_{\text{KL}} \left[ \pi(\cdot|x) \| \pi_{\text{ref}}(\cdot|x) \right] \right], \tag{2}$$

where $\beta > 0$ is the KL penalty coefficient. Several algorithms have been proposed to solve the above optimization, including PPO (Schulman et al., 2017; Ziegler et al., 2019), SLiC (Zhao et al., 2023), DPO (Rafailov et al., 2024), RSO (Liu et al., 2024b), and IPO (Azar et al., 2024). For a stable evaluation process, we use DPO in this work for preference alignment. For a given preference dataset $\mathcal{D}_{\text{p}}$[2], DPO uses the following loss function:

$$\mathcal{L}_{\text{DPO}}(\pi_\theta | \pi_{\text{ref}}, \mathcal{D}_{\text{p}}) = -\mathbb{E}_{(x, y_w, y_l) \sim \mathcal{D}_{\text{p}}} \left[ \log \sigma \left( \beta \log \frac{\pi_\theta(y_w|x)}{\pi_{\text{ref}}(y_w|x)} - \beta \log \frac{\pi_\theta(y_l|x)}{\pi_{\text{ref}}(y_l|x)} \right) \right] \tag{3}$$

**Reward Hacking** Reward model is not perfect due to its limited model size, limited training data, and distribution shift between training data and alignment prompts and responses (Eisenstein et al., 2023; Gao et al., 2023; Guo et al., 2024; Xiong et al.). Several works have been proposed to mitigate reward hacking. One line of works focus on observable artifacts such as length (Chen et al.; Dubois et al., 2024a; Shen et al., 2023b). Shen et al. (2023a) propose to enforce the consistency in reward model via data augmentation. To improve generalization, reward model ensembles can mitigate (but do not eliminate) reward hacking (Coste et al., 2023; Eisenstein et al., 2023; Ramé et al., 2024b). Reward hacking can also be mitigated during policy training with post-adjusted reward (Park et al., 2024) or with post-training model merge (Lin et al., 2023). We focus on improving the reward model by addressing reward hacking from a causal perspective.

**Causal Inference** Causal inference can be embedded in graphical model frameworks as a directed acyclic graph (DAG) $\mathcal{G} = (\mathcal{V}, \mathcal{E})$ with variables represented as nodes in $\mathcal{V}$ and causal relationship represented as a directed edge (Pearl, 2009; Lee et al., 2020) in $\mathcal{E}$. We say a random vector $X$ to be *faithful* with respect to a DAG $\mathcal{G} = (\mathcal{V}, \mathcal{E})$ if for any $i, j \in \mathcal{V}$, and any subset $S \subseteq \mathcal{V} \backslash \{i, j\}$,

$$X^i \perp X^j \mid X^S \Leftrightarrow i \text{ and } j \text{ are d-separated by } S \text{ under } \mathcal{G}, \tag{4}$$

where $X^i \perp X^j \mid X^S$ denotes $X^i$ and $X^j$ are independent conditional on $X^S$. The "d" in d-separation stands for dependence. Thus if $X^i$ and $X^j$ are d-separated relative to a set of variables

---

[1] We randomly flip response pairs and associated labels to remove positional bias.

[2] $\mathcal{D}_{\text{p}}$ can be $\mathcal{D}_{\text{hf}}$ or can be generated responses labeled by reward model as in Liu et al. (2024b)

$X^S$ in a directed graph, then they are independent conditional on $X^S$ in all probability distributions such a graph can represent. The definition of d-separation is as follows: suppose we are given a DAG $\mathcal{G}$; then, for two nodes $i, j \in \mathcal{V}$, a subset $S$ of $\mathcal{V}\backslash\{i, j\}$ d-connects $i$ and $j$ if there exists a path $L$ between $i$ and $j$ such that every collider in $L$ either belongs to $S$ or has a descendent in $S$, and no other node in $L$ belongs to $S$. If $S$ does not d-connect $i$ and $j$, then it d-separates $i$ and $j$. See Appendix A.4 for more details.

## 3 ROBUST REWARD MODEL TRAINING

We first formulate the reward model training in a causal framework, then we augment the reward model training data based on the causal rules.

### 3.1 CAUSAL FRAMEWORK

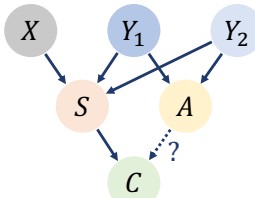

Figure 2: Causal graph of reward model. $X$ is the prompt. $Y_1, Y_2$ are two responses. $S$ is the contextual signal that depends on input prompt and two responses. $A$ is the context-free artifact that only depends on two responses. $C$ is the preference label. Traditional reward model cannot differentiate the two DAGs on whether there is a causal edge from $A$ to $C$. Our work uses the augmented dataset to eliminate the edge from $A$ to $C$.

We formulate a DAG $\mathcal{G}$ to model the causal relationships among different quantities (Figure 2). $X$ is the prompt, and $Y_1, Y_2$ are two responses. $S \in \mathbb{R}$ is the contextual signal that depends on input prompt and two responses. $A \in \mathbb{R}$ is the context-free artifact that only depends on two responses. $C \in \{0, 1\}$ is the preference label, where $C = 1$ means $Y_1$ is preferred over $Y_2$ and $C = 0$ means the other way around. We assume the distribution of $(X, Y_1, Y_2, S, A, C)$ to be faithful to the DAG. We assume the preference label $C$ can be captured by $S$ and $A$, which is to say $\mathbb{P}(C|X, Y_1, Y_2) = \mathbb{P}(C|S, A)$. We assume the $S$ to be the *sufficient statistic* (Lehmann & Casella, 2006) that captures the contextual effect that one response fulfills the need of the prompt better than the other. We assume $A$ to the *sufficient statistic* that captures the context-free artifacts that only depend on two responses. Such artifacts can include length, format (bold faces, bullet points, markdown, etc), and certain patterns ($n$-grams such as "Sure, here is the response:"). In traditional reward model training, the model may hack the patterns in $(Y_1, Y_2)$. Suppose 80% of winning responses to be longer, then the reward model can get 80% accuracy by just counting the number of tokens. Formally, we construct two hypothesis:

- $\mathcal{H}_0$: there is no causal edge from $A$ to $C$.
- $\mathcal{H}_1$: there is a causal edge from $A$ to $C$.

**Proposition 3.1.** *In traditional reward model training, $\mathcal{H}_0$ and $\mathcal{H}_1$ are not always distinguishable.*

*Proof.* As an example of the hypotheses being indistinguishable, let's consider a special case of $A$ and $S$ being perfectly correlated. More formally, assume $S = s(X, Y_1, Y_2) + \epsilon_s$ with certain non-linear function $s$ and $\epsilon_s \sim N(0, \sigma_s)$, and similarly $A = a(X, Y_1, Y_2) + \epsilon_a$ with non-linear function $a$ and $\epsilon_a \sim N(0, \sigma_a)$. Suppose $\mathbb{P}(C = 1|X, Y_1, Y_2) = \sigma(\beta_s S + \beta_a A + \alpha + \epsilon_c)$ with constants $\beta_s, \beta_a, \alpha \in \mathbb{R}$ and random error $\epsilon_c \perp (S, A)$. If $\beta_a = 0$, then $\mathcal{H}_0$ is true. If $\beta_a = 1$, then $\mathcal{H}_1$ is true. In extreme case that $Corr(S, A) = 1$, then $A = \beta_{as} S + \alpha_a$ for some constants $\alpha_a \in \mathbb{R}$ and $\beta_{as} \in \mathbb{R}^+$. Then the model cannot tell if $\beta_a = 0$ or not. This is because when $\beta_a = 0$, we can still reparametrize it as $\mathbb{P}(C = 1|X, Y_1, Y_2) = \sigma((\beta_s - \beta_{as})S + A + (\alpha - \alpha_a) + \epsilon_c)$. □

The desired behavior of a reward model is to determine the preference label $C$ ignoring the artifact $A$, which corresponds to $\mathcal{H}_0$. To achieve that, we can utilize two d-separation relationships of the DAG $\mathcal{G}$.

- **R1**: Under $\mathcal{H}_0$, $A$ and $C$ are d-separated by $(Y_1, Y_2)$, thus $A \perp C \mid (Y_1, Y_2)$.
- **R2**: Under $\mathcal{H}_0$, $A$ and $C$ are d-separated by $S$, thus $A \perp C \mid S$.

## 3.2 Data augmentation

To fix the issue mentioned in Proposition 3.1, we can effectively utilize **R1&R2**. In particular, we propose to augment data with by adding the permuted pairs of generated responses.

**Possible Combinations**    Given the dataset of triplets $\mathcal{D}_{\text{hf}} = \{t^{(i)}\}_{i=1}^N$ with $t^{(i)} = (x^{(i)}, y_w^{(i)}, y_l^{(i)})$, we can first expand the dataset as $\tilde{\mathcal{D}}_{\text{hf}} = \{t^{(i)}, t^{(\sigma_1(i))}, t^{(\sigma_2(i))}\}_{i=1}^N$, where $\sigma_1 : [N] \to [N]$ and $\sigma_2 : [N] \to [N]$ are two different invertible permutation functions randomly sampled from permutation group $S_N$. In practice, we can shuffle the dataset twice to achieve $\sigma_1$ and $\sigma_2$. There are in total $3 \times \binom{6}{2} = 45$ possible $(x, y_1, y_2)$ unordered triplets from each element in $\tilde{\mathcal{D}}_{\text{hf}}$. This is because there are 3 possible prompts with 2 choices among 6 responses and we treat $(x, y_1, y_2)$ and $(x, y_2, y_1)$ as the same one.

**Preference Labels**    For the unordered triplet, we can set the preference rule based on the DAG $\mathcal{G}$. We say response $y$ is *contextual* to $x$ if they are from the same triplet in $\mathcal{D}_{\text{hf}} = \{x^{(i)}, y_w^{(i)}, y_l^{(i)}\}_{i=1}^N$. For example, $y_w^{(i)}$ and $y_l^{(i)}$ are contextual to $x^{(i)}$, but $y_w^{(j)}$ and $y_l^{(j)}$ are not contextual to $x^{(i)}$ for $j \neq i$. Then for $(x, y_1, y_2)$, we have the following rules:

- if both $y_1$ and $y_2$ are contextual to $x$, we set the winning one in $\mathcal{D}_{\text{hf}}$ as winner.
- if only one of $y_1$ and $y_2$ is contextual to $x$, we set the contextual one as winner.
- if neither $y_1$ nor $y_2$ is contextual to $x$, we set the preference label as "Tie".

Here we assume that $y_l^{(i)}$ is still an acceptable response for $x^{(i)}$ because it is usually generated by a language model conditional on $x^{(i)}$.

**Augmented Triplets**    From **R1**, we can fix $(Y_1, Y_2)$ and vary $X$ to perturb $C$. From **R2**, we can fix $C$ by picking a contextual (prompt, response) pair $(X, Y_1)$ and another non-contextual response $Y_2$. Then we set $Y_1$ as winning response and vary losing response $Y_2$ to perturb $A$. We can see the augmented datasets derived from the above two rules cover all possible $(x, y_1, y_2)$ unordered triplets generated from $\tilde{\mathcal{D}}_{\text{hf}}$. For simplicity, we select the ones with prompt $x^{(i)}$, which provides us with the following additional augmented triplets[3]:

$$
\left. \begin{array}{l}
(x^{(i)}, y_w^{(i)}, y_w^{(\sigma_1(i))}) \to \text{chosen} = y_w^{(i)} \\
(x^{(i)}, y_w^{(i)}, y_w^{(\sigma_2(i))}) \to \text{chosen} = y_w^{(i)} \\
(x^{(i)}, y_w^{(i)}, y_l^{(\sigma_1(i))}) \to \text{chosen} = y_w^{(i)} \\
(x^{(i)}, y_w^{(i)}, y_l^{(\sigma_2(i))}) \to \text{chosen} = y_w^{(i)} \\
(x^{(i)}, y_l^{(i)}, y_w^{(\sigma_1(i))}) \to \text{chosen} = y_l^{(i)} \\
(x^{(i)}, y_l^{(i)}, y_w^{(\sigma_2(i))}) \to \text{chosen} = y_l^{(i)} \\
(x^{(i)}, y_l^{(i)}, y_l^{(\sigma_1(i))}) \to \text{chosen} = y_l^{(i)} \\
(x^{(i)}, y_l^{(i)}, y_l^{(\sigma_2(i))}) \to \text{chosen} = y_l^{(i)}
\end{array} \right\} \text{Non-contextuals}
\qquad
\left. \begin{array}{l}
(x^{(i)}, y_w^{(\sigma_1(i))}, y_l^{(\sigma_1(i))}) \to \text{Tie} \\
(x^{(i)}, y_w^{(\sigma_2(i))}, y_l^{(\sigma_2(i))}) \to \text{Tie} \\
(x^{(i)}, y_w^{(\sigma_1(i))}, y_w^{(\sigma_2(i))}) \to \text{Tie} \\
(x^{(i)}, y_w^{(\sigma_1(i))}, y_l^{(\sigma_2(i))}) \to \text{Tie} \\
(x^{(i)}, y_w^{(\sigma_2(i))}, y_l^{(\sigma_1(i))}) \to \text{Tie} \\
(x^{(i)}, y_l^{(\sigma_1(i))}, y_l^{(\sigma_2(i))}) \to \text{Tie}
\end{array} \right\} \text{Neutrals}
\tag{5}
$$

---

[3]We show a sample Python code in Appendix A.2.

"Non-contextuals" set the contextual response as chosen and non-contextual one as rejected. "Neutrals" set both non-contextual responses as tie. With these, we have the following claim:

**Proposition 3.2.** *If the reward model is trained with $\mathcal{D}_{hf}$ and augmented triplets in Equation 5, there is no causal edge from $A$ to $C$ in DAG $\mathcal{G}$.*

*Proof.* We can prove this by contradiction. If there is a causal edge from $A$ to $C$, then the conditional independence relations $A \perp C \mid (Y_1, Y_2)$ and $A \perp C \mid S$ do not hold, which contracts to the triplets constructed on "Non-contextuals" and "Neutrals". □

### 3.3 CONNECTION TO EXISTING WORKS

**ODIN (Chen et al.)**    ODIN decomposes reward into additive format of a quality one and a length one. During learning, it enforces the disentanglement between the quality reward and the response length and encourages the correlation between the length reward and the response length. We claim that this is a special case of our causal modelling with single observed artifact $A$ as length, because the disentangle learning is a necessary condition of the conditional independence between $C$ and $A$ given the data. Our framework is more general and can go beyond single and observed artifact.

**Length-controlled AlpacaEval-2 (Dubois et al., 2024a)**    This work improves the original version of AlpacaEval-2 by conditioning on the length through Controlled Direct Effect (VanderWeele, 2011). It adds length as a variable in the logistic regression to predict the preference. Effectively, it learns the residual part that cannot be explained by the length. In our framework, we directly learn the residual part that is orthogonal to the artifacts, which is the length in length-controlled AlpacaEval-2. Thus the two methods are equivalent, and our approach can go beyond single artifact and be extended to unobserved artifacts.

**Length-controlled DPO (Park et al., 2024)**    This work adds a length penalty in the RLHF objective (Equation 2). It serves as a post-hoc reward adjustment to mitigate the length bias during policy optimization. The idea for removing the lengthy bias using a length reward is the same as ODIN, but they don't have the correlation penalty and the additional hyperparameter introduced can add more complexity into policy optimization. In comparison, our work directly learns a artifact-free reward model so we do not need an explicit length adjustment factor in the alignment algorithm designs.

**Contrast Instructions (Shen et al., 2023a)**    This work shows the issues of reward models on the instruction and response consistencies when switching instruction or response to another similar one. It proposes a data augmentation training approach and retrieval-augmented inference technique to improve the consistencies of reward models. On contrary, by considering all possible combinations of $(x, y_1, y_2)$ across different examples, our approach uses the organic data from the dataset, which can effectively eliminate the artifacts existing in the dataset.

## 4 EXPERIMENTS

In this section, we conduct reward modeling and apply the trained reward to downstream alignment tasks to verify the effectiveness of the proposed method. For deeper understanding, we also conduct analysis on reward model training data, aligned policies, and perturbed reward model training data.

### 4.1 SETTINGS

**Training Set**    We study RRM using the preference dataset curated by RLHFlow[4] (Dong et al., 2024), which has been used to train a series of strong open-source preference models as evaluated by the Reward-Bench (Lambert et al., 2024). The dataset consists of 700K preference pairs, which is a mixture of HH-RLHF (Bai et al., 2022a), SHP (Ethayarajh et al., 2022), HelpSteer (Wang et al., 2023), PKU-SafeRLHF (Ji et al., 2024), UltraFeedback (Cui et al., 2023), UltraInteract (Yuan et al., 2024), Distilabel-Capybara (Daniele & Suphavadeeprasit, 2023), and Distilabel-Orca (Lian et al., 2023). We list the data sources and number of examples in Table 1. The authors of the original

---

[4]https://huggingface.co/datasets/RLHFlow/pair_preference_model_dataset

paper delete the samples with similar scores when the scores are available because when the model is well calibrated, these samples are more likely to mislabelled. Thus the total number is smaller than the sum of each individual datasets.

| Source | Number of Examples |
|---|---|
| HH-RLHF-Helpful[5] (Bai et al., 2022a) | 115,396 |
| SHP[6] (Ethayarajh et al., 2022) | 93,301 |
| HelpSteer[7] (Wang et al., 2023) | 37,131 |
| PKU-SafeRLHF[8] (Ji et al., 2024) | 26,874 |
| UltraFeedback[9] (Cui et al., 2023) | 340,025 |
| UltraInteract[10] (Yuan et al., 2024) | 161,927 |
| Distilabel-Capybara[11] (Daniele & Suphavadeeprasit, 2023) | 14,811 |
| Distilabel-Orca[12] (Lian et al., 2023) | 6,926 |

Table 1: Composition of reward model training dataset

**Reward Model Training Details**  We first train a pairwise ranking reward model (RM) from Gemma-2-9b-it. With the augmentation illustrated in Equation 5, we can get 14X additional examples, most of which can be too easy for RM to learn. To reduce the augmented data size, we first conduct inference on random 50% of the augmented data using the trained RM, and leave the examples with $|\hat{\mathbb{P}}(A \succ B) - \mathbb{P}^*(A \succ B)| \geq 0.2$, where $\hat{\mathbb{P}}(A \succ B)$ is winning probability calculated by RM and $\mathbb{P}^*(A \succ B)$ is the ground truth probability[13]. We get 2.4M training examples by merging the filtered augmented data and original RM training data. Then we use the same training recipe to get the robust reward model (RRM). We train the reward models for 1 epoch using AdamW (Loshchilov, 2017) optimizer with learning rate 1e-6 and batch size 128[14].

**Policy Model Training Details**  We train DPO policies using the on-policy responses generated by Gemma-2-9b-it and labeled by RM and RRM, respectively. We use the prompt set from the UltraFeedback dataset to generate 5 responses per prompt. Then, we compare all $\binom{5}{2}$ pairs and pick the best-worst response pairs to align the DPO policy following (Pace et al., 2024; Dong et al., 2024). We train the policies for 2 epochs at most using AdamW (Loshchilov, 2017) optimizer with learning rate 2e-7 and a global batch size of 128, where the batch size follows Dong et al. (2024) and the learning rate is decided by grid search.

**Evaluation Metrics**  We evaluate the quality of reward model from two perspectives: the accuracy on Reward-Bench (Lambert et al., 2024) and the quality of policies induced by the reward model. For policies induced by the reward model, we consider two variants: 1. Best-of-N (BoN) policy and 2. aligned DPO policy. Our main focus is for open-ended generation and we use MT-Bench (Zheng et al., 2024) and AlpacaEval-2 (Dubois et al., 2024b) to evaluate.

## 4.2 MAIN RESULTS

**Reward Model Accuracy**  The test accuracies on Reward-Bench are reported in Table 2. RRM improves "Chat Hard" and "Safety" by a clear margin but sacrifices the "Reasoning". Regarding "Reasoning", we hypothesize that math and coding are less affected by the non-contextual artifacts and we may use other rewards than an LLM because those are objectives like golden answers. On average, RRM improves RM by an absolute 3.54% accuracy gain.

---

[5]https://huggingface.co/datasets/RLHFlow/HH-RLHF-Helpful-standard
[6]https://huggingface.co/datasets/RLHFlow/SHP-standard
[7]https://huggingface.co/datasets/RLHFlow/Helpsteer-preference-standard
[8]https://huggingface.co/datasets/RLHFlow/PKU-SafeRLHF-30K-standard
[9]https://huggingface.co/datasets/RLHFlow/UltraFeedback-preference-standard
[10]https://huggingface.co/datasets/RLHFlow/UltraInteract-filtered-standard
[11]https://huggingface.co/datasets/RLHFlow/Capybara-distibalel-Filter-standard
[12]https://huggingface.co/datasets/RLHFlow/Orca-distibalel-standard
[13]The ground truth probability is 1 if A is preferred over B, 0 if B is preferred over A, and 0.5 if they tie.
[14]We found 1 epoch is best for reward model training and we pick the best hyperparameter by grid search.

| Model | Chat | Chat Hard | Safety | Reasoning | Average |
|-------|------|-----------|--------|-----------|---------|
| RM | **97.77** | 51.54 | 78.54 | **94.58** | 80.61 |
| RRM | 96.51 | **65.57** | **83.90** | 90.62 | **84.15** |

Table 2: Comparison of test accuracy of Reward-Bench. RRM shows improvement upon RM on Chat Hard and Safety with an average 3.54% improvement of accuracy.

**Policies Induced by Reward Models**    We investigate the quality of reward models by evaluating the aligned policies. To study the effect of adding "Neutrals" in Equation 5, we also train a reward model without augmented neutrals (-Neutrals). The results are summarized in Table 3. As expected, ODIN (Chen et al.)[15] shows shorter responses than RM and RRM since it explicitly disentangles the length from quality. RRM shows the best performance on MT-Bench first turn and AlpacaEval-2 over ODIN and RM, with shorter responses generated than RM, suggesting it effectively controls the length as one of the artifact. The added "Neutrals" have slight improvements on first-turn MT-Bench and AlpacaEval-2.

| Reward | Policy | MT-Bench[16] | | | AlpacaEval-2 | | |
|--------|--------|--------|--------|-------------|-------------|-------------|-------------|
| | | T1 ($\uparrow$) | T2 ($\uparrow$) | Overall ($\uparrow$) | LC (%) ($\uparrow$) | WR (%) ($\uparrow$) | Length ($\downarrow$) |
| RM | BoN (N=8) | - | - | - | 36.87 | 50.14 | 3072 |
| RRM | BoN (N=8) | - | - | - | **47.68** | **53.19** | **1770** |
| RM | BoN (N=64) | - | - | - | 40.52 | 57.62 | 2992 |
| RRM | BoN (N=64) | - | - | - | **62.82** | **63.03** | **1770** |
| RM | DPO | 8.02 | 6.33 | 7.27 | 33.46 | 41.07 | 2416 |
| ODIN | DPO | 8.66 | 8.13 | 8.39 | 48.29 | 37.13 | **1559** |
| RRM | DPO | **8.70** | 7.87 | 8.31 | **52.49** | **43.31** | 1723 |
| -Neutrals | DPO | 8.65 | **8.21** | **8.44** | 51.73 | 43.24 | 1722 |

Table 3: Comparison among different reward models on various aligned policies. T1 and T2 stand for the first and second turn of the conversation, respectively. WR stands for win-rate against GPT-4. LC stands for length-controlled win-rate. Length is the average number of characters in the generated responses. RRM shows quality improvements over ODIN and RM with shorter responses than RM. Dropping augmented neutral examples slightly hurt the quality.

## 4.3    LENGTH ANALYSIS

To further understand the artifacts learned in reward model, we take length as an example to analyze the reward model training data and aligned policy.

**Length distribution of training data**    We study the length (number of tokens) distribution of reward model training datasets. Length is one common artifact that shows bias on both policy training and evaluation. We hypothesize that the bias can possibly come from the reward model training data. The one used in training RM is not well calibrated and the chosen responses are longer on average (Figure 3a) and by frequency (Figure 3c). On contrary, the RRM training data is better calibrated with length more balanced between chosen and rejected responses in each length bin (Figure 3b and 3c). We further provide length analysis for each data source in Appendix A.1.

**Length distribution of policies**    To understand the lengthy bias learned in various policies, we also study the length distribution of generated responses on AlpacaEval-2's (Dubois et al., 2024a) prompts (Figure 4). We observe that the policies induced by RRM generate shorter responses than RM, which implies the correction of lengthy bias by RRM.

---

[15]Training details in Appendix A.3.

[16]we do not evaluate BoN policies on MT-Bench because it involves multi-turn.

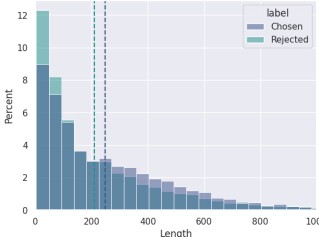 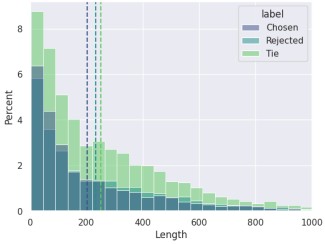 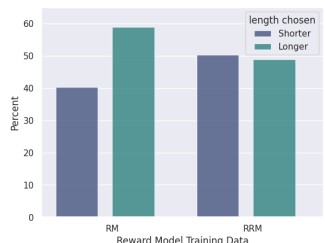

(a) Histogram of response lengths in RM training data.

(b) Histogram of response lengths in RRM training data.

(c) Percentage of chosen responses being longer or shorter in RM and RRM traininng data.

Figure 3: Distribution of response lengths on reward model training datasets. (a) the RM training data has longer chosen responses on average and not well calibrated (large percent deviation in left two bins between chosen and rejected) (b) the RRM training data is well calibrated and the average length of the chosen responses is even shorter than rejected. Additional neutral triplets can further calibrated the model. (c) Around 60% of chosen responses are longer in RM training data. On contrary, the lengths of chosen responses are more balanced in RRM training data.

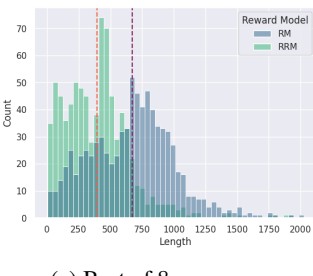 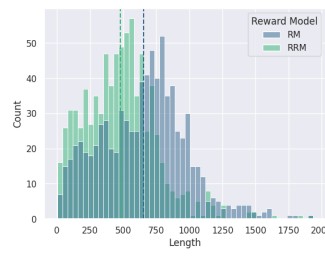 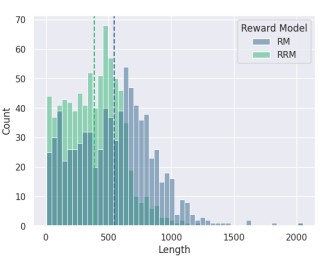

(a) Best of 8 responses

(b) Best of 64 responses

(c) DPO policy

Figure 4: Distribution of response lengths on AlpacaEval-2 prompts of various policies induced by RM and RRM, average length is marked by the dashed line. All policies show a lengthy bias towards longer responses for RM comparing with RRM.

## 4.4 DELIBERATELY DESIGNED ARTIFACTS

**Artifacts** To verify that our proposed method is able to eliminate artifacts, we artificially added an artifact to the chosen responses in reward model training data. More specifically, we add prefix "*Sure, here is the response:* " to the chosen responses with probability 0.1. We train an RM and RRM on the modified reward model training data, respectively.

To test the effect of reward model on the policy model, we first sample $N$ responses from Gemma-2-9b-it model using the AlpacaEval-2 prompts. Then we add the same type of artifact to each response with probability $p_a = p$, where $p \in \{0.05, 0.1, 0.2, 0.5\}$. Under this setting, RM trained on the artifact-added data would prefer responses with the artifacts since the chosen responses come with artifacts, even if the responses may contain low-quality answer. RRM is expected to be more robust to the artifact. To verify this, we construct BoN policies using RM and RRM, respectively.

As expected, Figure 5 shows that after adding the artifacts, the BoN policies induced by RRM are more robust than RM to artifacts injected in the responses.

## 5 RELATED WORKS

**RLHF algorithms** The first RLHF framework (Stiennon et al., 2020) is based on the proximal policy optimization (PPO) algorithm, which was first popularized by Christiano et al. (2017) and further developed by Bai et al. (2022a); Ouyang et al. (2022). However, getting PPO work is challenging especially in the era of LLMs (Choshen et al.; Engstrom et al., 2020). In recognition of this issue, another line of works propose direct alignment algorithms, where notable examples include

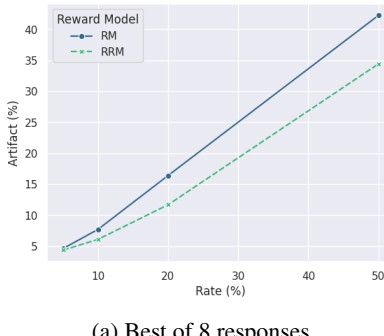 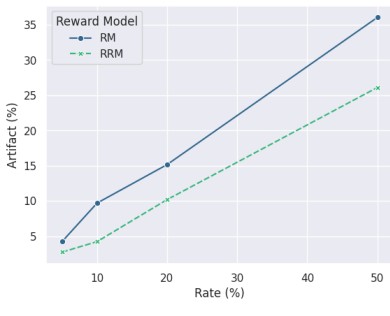

(a) Best of 8 responses        (b) Best of 64 responses

Figure 5: Proportion of BoN generated responses with artifact versus the rate of injected artifact. For each policy, we first sample $N$ ($N = 8$ or $64$) responses on AlpacaEval-2 prompts, then prepend "Sure, here is the response: " before each response with probability (Rate) 5%, 10%, 20%, 50%, respectively. Then we compute the proportion of BoN responses that have the above artifact (Artifact). The BoN policies induced by RRM are more robust to artifacts injected in the responses, suggesting that the proposed approach enables the model to focus more on the contextual signals instead of context-free artifacts in the reward model training data.

SLiC (Zhao et al., 2023), DPO (Rafailov et al., 2024), IPO (Azar et al., 2024), KTO (Ethayarajh et al., 2024), ORPO (Hong et al., 2024), SimPO (Meng et al., 2024), and DRO (Richemond et al., 2024). These algorithms directly optimize a supervised target to optimize the policy model without constructing a reward model first, hence the name direct alignment algorithms. However, these algorithms learning from a fixed dataset are offline and often off-policy without further exploration of the environment. RSO (Liu et al., 2024b) emphasizes the importance of reward model and fixes the distribution shift problem to improve the DPO training, followed by list-wise alignment (Liu et al., 2024a) and the online (iterative) training frameworks (Xiong et al.; Guo et al., 2024; Calandriello et al.). Alternatively, there is also a line of work based on the best-of-n sampling, such as RAFT (Dong et al.), BOND (Sessa et al., 2024), BoNBoN alignment (Gui et al., 2024). These algorithms leverage a reward model to rank the generated responses and distill knowledge from the best responses. Our approach can benefit RLHF algorithms relying on a reward model.

**Reward Models & Reward Hackings** Building a superhuman/unbiased reward model is vital for training better chat assistants since it could affect the upper bound of the policies' capabilities in the online preference optimization (Wang et al., 2024a; Bai et al., 2022b). Multi-objective rewards (Wang et al., 2024b), RLHF-workflow (Dong et al., 2024), and RMBoost (Shen et al., 2024) are proposed to train more capable reward models. While revealed by Denison et al. (2024); Zhang et al. (2024), reward models are easily hacked by different pattern in different scenario, e.g., length (Singhal et al., 2023) and sycophancy. Recent works employ the model merging (WARP (Ramé et al., 2024a) and WARM (Ramé et al., 2024b)), and hacking reward decomposition (ODIN (Chen et al.)) to mitigate the hackings in online RLHF. Generative reward models can provide more detailed preference analysis (Yan et al., 2024). For the most accurate reward signal, one can also use verifiable answers in certain domain like math (Xiong et al., 2024). Most model-based methods failed to distinguish between preferences driven by the prompt and context-free artifacts. Our RRM is more advanced in removing the artifacts.

## 6 CONCLUSION

In this paper, we identified a key problem in the current reward training methodology: its inability to differentiate between contextual signals and context-free artifacts. Using a causal framework, we explained this effect and improved reward model training by introducing a data augmentation approach derived from the framework. Our theoretical analysis and extensive empirical results demonstrated that the proposed techniques effectively enhance both the test accuracy of the reward model and the quality of the policies it induces. Future work will explore filtering augmented pairs and matching artifacts when constructing response pairs, further refining the training process.

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

# A   APPENDIX

## A.1   ADDITIONAL LENGTH ANALYSIS OF REWARD MODEL TRAINING DATASETS

In this section, we show the length distribution of chosen and rejected responses from each individual data source in Figure 6. HH-RLHF, SHP, HelpSteer, UltraFeedback show bias towards rejected responses as the first length bin. SHP, HelpSteer, and UltraFeedback have longer chosen responses than rejected ones on average.

## A.2   DATA AUGMENTATION PYTHON CODE

In Algorithm 1, we show a sample code of data augmentation in Python. We expect each element in data contains "context", "response_w", "response_l". We use "neutral" to indicate if the label should be "Tie".

## A.3   TRAINING DETAILS FOR ODIN

We use the same loss as described in Chen et al.. We train Gemma-2-9b-it for 1 epoch on the same data we used for RM. AdamW is our optimizer and the learning rate is set to 2e-6 with cosine scheduler. We use Flash-Attention to accelerate the training while applying the Deepspeed Zero-Stage 3 to get batch size 16 on each GPU (the global batch size is 128) to make sure the calculation of the Pearson correlation between the head value and the length of the responses is stable.

## A.4   PRELIMINARIES IN CAUSAL INFERENCE

We list a few critical concepts in this section. For information, we refer the readers to Lauritzen (1996) and Pearl (2009).

**DAGs and d-separation**   A DAG is a set of vertices and a set of directed edges (arrows) that connect pairs of these vertices. The causal modeling connects a DAG with Markov condition via a graphical relation called *d-separation* (Pearl, 2009). D-separation is a relation among three disjoint sets of vertices in a directed graph. D-separation and Markov condition connect DAGs and probability distribution. By *faithfulness* assumption, the d-separation in a DAG is equivalent to conditional independence in distribution.

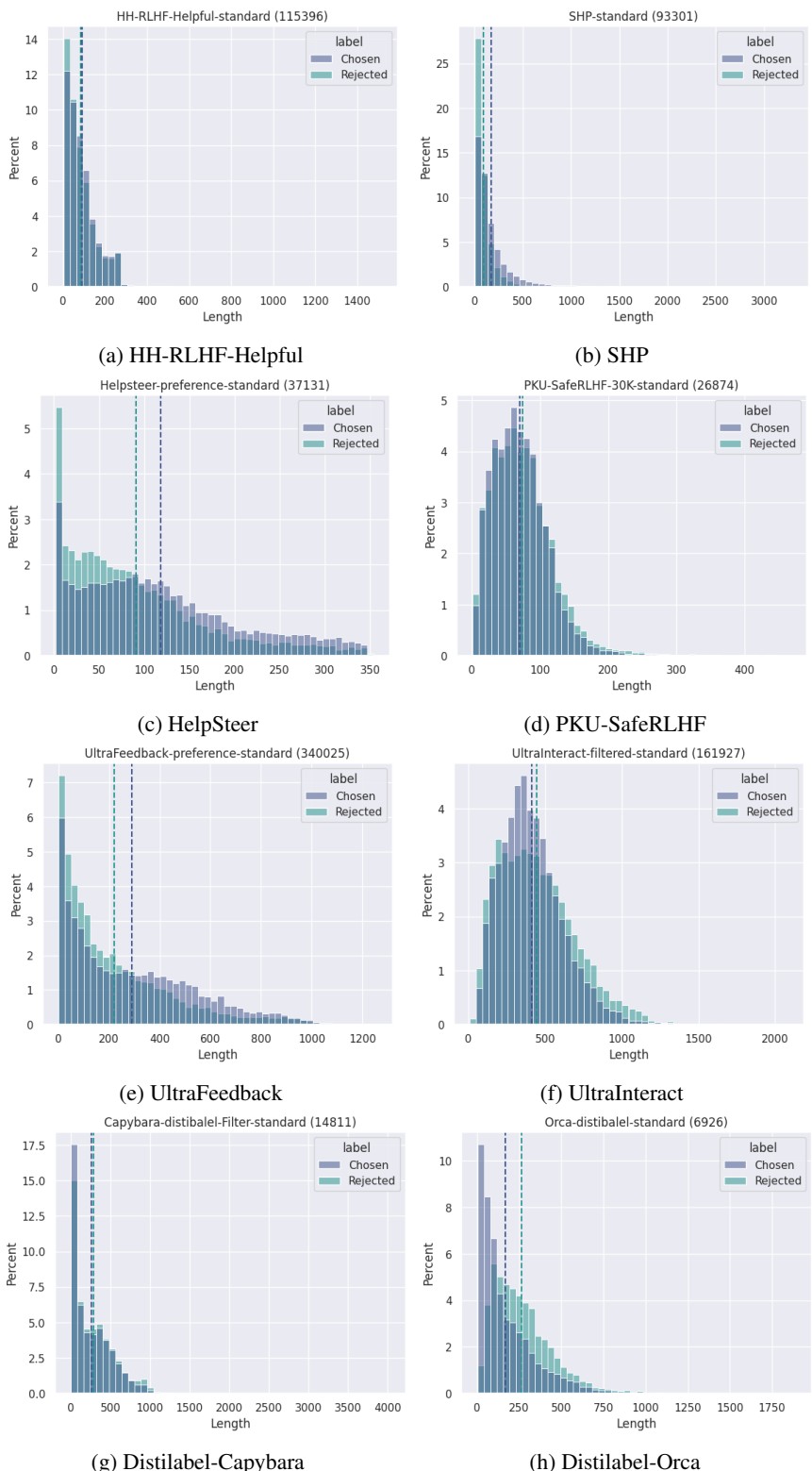

Figure 6: Distribution of response lengths on each individual source of reward model training data. SHP, HelpSteer, and UltraFeedback show significant lengthy bias showing longer responses in chosen. They also dominate the training dataset, accounting for more than a half.

---

**Algorithm 1** Example Python Code for Data Augmentation

---

```python
def get_augmented(data: List[Dict[str, Any]]) -> List[Dict[str, Any]]:
  data_i = data
  data_j = data_i.copy()
  random.shuffle(data_j)
  data_k = data_j.copy()
  random.shuffle(data_k)
  for ex_i, ex_j, ex_k in zip(data_i, data_j, data_k):
    xi = ex_i['context']
    xj = ex_j['context']
    xk = ex_k['context']
    ywi = ex_i['response_w']
    ywj = ex_j['response_w']
    ywk = ex_k['response_w']
    yli = ex_i['response_l']
    ylj = ex_j['response_l']
    ylk = ex_k['response_l']
    # xi_ywi_ywj
    yield {
        "context": xi,
        "response_w": ywi,
        "response_l": ywj,
        "neutral": False
    }
    # xi_ywi_ywk
    yield {
        "context": xi,
        "response_w": ywi,
        "response_l": ywk,
        "neutral": False
    }
    # fill in all other augmented triplets ...
    # xi_ywk_ylj
    yield {
        "context": xi,
        "response_w": ywk,
        "response_l": ylj,
        "neutral": True
    }
    # xi_ylj_ylk
    yield {
        "context": xi,
        "response_w": ylj,
        "response_l": ylk,
        "neutral": True
    }
```

---

**The causal Markov condition**   The Causal Markov assumption assumes that a variable $X$ is independent of every other variable (except $X$'s effects) conditional on all of its direct causes. With this, a DAG defines a set of distributions of the form

$$p(y_1, ..., y_k) = \prod p(y_j | \text{parents}(y_j))$$

**Counterfactuals**   Consider two variables $X$ and $Y$. We will call $X$ the "treatment". We call $Y$ the "outcome". For a given subject we see $(X_i, Y_i)$. What we don't see is what their value of $Y_i$ would have been if we changed their value of $X_i$. This is called *counterfactual*. Suppose that $X$ is a binary variable that represents some treatment. So $X = 1$ means the subject was treated and $X = 0$ means the subject was not treated. Let $Y_1$ denote the outcome if the subject is treated. Let $Y_0$ denote the response if the subject is not treated. Then

$$Y = XY_1 + (1 - X)Y_0$$

If we treat a subject, we observe $Y_1$ but not $Y_0$. The unobserved variable is called a *counterfactual*. The variables $(Y_0, Y_1)$ are also called *potential outcomes*. We define *mean treatment effect* as

$$\theta = \mathbb{E}(Y_1) - \mathbb{E}(Y_0) = \mathbb{E}(Y|\text{set } X = 1) - \mathbb{E}(Y|\text{set } X = 0)$$

## A.5  ADDITIONAL RESULTS WITH GEMMA-2-2B-IT

To further verify effectiveness of our approach, we train Gemma-2-2b-it reward model (RM) and robust reward model (RRM), respectively. Table 4 shows the results on the reward bench. RRM again shows improvement on the Chat Hard. It shows some regression on Safety and Reasoning, where we hypothesis that some context-free nature of reasoning and safety makes the RRM perform worse. Overall, RRM shows positive effect. We also test the reward models on AlpacaEval2 using best-of-n policies. The results are shown in Table 5. We use the trained reward model to rank the responses generated from the Gemma-2-9b-it model, and observe consistent gains on RRM over RM.

| Model | Chat | Chat Hard | Safety | Reasoning | Average |
|-------|------|-----------|--------|-----------|---------|
| RM | 96.49 | 42.54 | 72.70 | **72.30** | 71.01 |
| RRM | **97.21** | **49.01** | **72.71** | 70.08 | **72.25** |

Table 4: Comparison of test accuracy of Reward-Bench. RRM shows improvement upon RM on Chat and Chat Hard with an average 1.75% improvement of accuracy.

| Reward | Policy | AlpacaEval-2 | | |
|--------|--------|--------------|--------------|-------------|
| | | LC (%) ($\uparrow$) | WR (%) ($\uparrow$) | Length ($\downarrow$) |
| RM | BoN (N=8) | 44.21 | 45.83 | 2264 |
| RRM | BoN (N=8) | **54.37** | **47.19** | **1749** |
| RM | BoN (N=64) | 47.32 | 52.23 | 2316 |
| RRM | BoN (N=64) | **60.42** | **56.50** | **1870** |

Table 5: Comparison among different reward models on various aligned policies. WR stands for win-rate against GPT-4. LC stands for length-controlled win-rate. Length is the average number of characters in the generated responses. RRM shows quality improvements with shorter responses over RM.

## A.6  ADDITIONAL ANALYSIS WITH MIXED ARTIFACTS

To investigate the effect of RRM on mixed artifacts, we conduct an additional experiment as follows:

1. With $p = 0.1$, wrap the whole chosen response with "**" as bold-face.
2. After the above step, with $p = 0.1$, append emoji 😊.
3. Train RM and RRM on the above dataset.

To test the effect of reward model on the policy model, we first sample $N$ responses from Gemma-2-9b-it model using the AlpacaEval-2 prompts. Then we add emoji 😊 to each response with probability $p_a = p$, where $p \in \{0.05, 0.1, 0.2, 0.5\}$. Under this setting, RM trained on the artifact-added data would prefer responses with the artifacts since the chosen responses come with artifacts, even if the responses may contain low-quality answer. RRM is expected to be more robust to the artifact. To verify this, we construct BoN policies using RM and RRM, respectively.

As expected, Figure 7 shows that after adding the artifacts, the BoN policies induced by RRM are more robust than RM to artifacts injected in the responses.

## A.7  DISCUSSION ON DATA FILTERING STRATEGIES

In this work, we assume the preference labels should be purely controlled by the prompt dependent signal. However, there can be cases such that prompt-independent signals can contribute to the

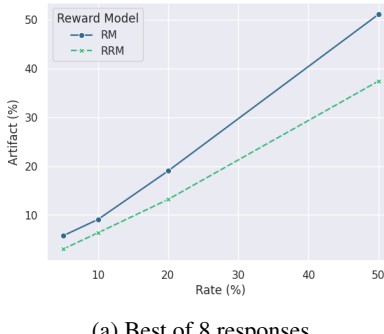 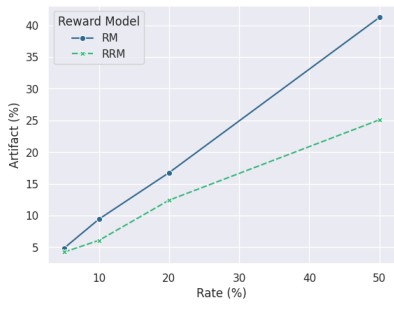

(a) Best of 8 responses            (b) Best of 64 responses

Figure 7: Proportion of BoN generated responses with emoji versus the rate of injected emoji. For each policy, we first sample $N$ ($N = 8$ or $64$) responses on AlpacaEval-2 prompts, then append emoji after each response with probability (Rate) 5%, 10%, 20%, 50%, respectively. Then we compute the proportion of BoN responses that have the above artifact (Artifact). The BoN policies induced by RRM are more robust to artifacts injected in the responses, suggesting that the proposed approach enables the model to focus more on the contextual signals instead of context-free artifacts in the reward model training data.

preference label. For example, a responsible AI should be always safe and cares about the diversity. In the data augmentation, we sometimes use rejected response as new chosen in "Non-contextuals". See the last four triplets in "Non-contextuals" of Equation 5. For "Neutrals", we also assume 0.5 winning probability of a non-contextual response pair. These treatment may cause the unwanted behavior of AI if we use unsafe response as the new chosen or assigning winning probability of 0.5 on a pair of (safe, unsafe) responses.

To address this, we have a few treatments that can be applied in future works:

- Use a trained safety pointwise or Bradley-Terry model to filter out triplets that has low safety scores on the chosen responses.
- Use AI feedback such as Constitutional AI (Bai et al., 2022c) to ensure the augmented triplets have high quality chosen responses and consistent preference according to certain non-contextual rules. The rules can include safety, style, factuality.

## B   ETHICS STATEMENT

Our research adheres to the ethical guidelines. Our work aims to mitigate reward hacking in RLHF, contributing to the development of more reliable AI systems that better align with human preferences. Our data augmentation process explicitly addresses and mitigates common forms of bias, thus reducing the potential for harm in practical applications of AI systems. Our model was designed with fairness in mind, particularly in avoiding biases related to response length and format, which can unfairly influence AI decision-making.

## C   REPRODUCIBILITY STATEMENT

All code used for training the reward models (RM and RRM) and for running the experiments described in this paper will be made publicly available upon publication. This includes the implementation of the data augmentation pipeline, the reward model training process, and policy alignment. The datasets used in our experiments are publicly available. We provide the complete set of hyperparameters used for training the models, including learning rates, batch sizes, and other optimization settings. The evaluation approaches are also public available and can be fully reproduced.

