# OpenReview forum: "RRM:  Robust Reward Model Training Mitigates Reward Hacking"
_ICLR.cc/2025/Conference — ICLR 2025 Poster_

### Official Review · Reviewer_1CBu · 2024-10-29

**Soundness:** 1
**Presentation:** 3
**Contribution:** 2
**Rating:** 5
**Confidence:** 3

**Summary:**

This paper proposes a robust reward model (RRM) training method to mitigatge reward hacking by utilizing causal inference framework. A causal graph for human preference modeling is introduced to enable the model to distinguish contextual preference signals and context-free artifacts. The training data is augmented by re-organizing the (prompt, positive, negative) triplet guided by the causal inference framework. Experiments show that the proposed approach can filter out undesirable artifacts in reward model training and yields a more robust reward model. Specifically, the reward model is trained on Gemma-2-9b-it, and RRM improves RM by an absolute 3.54% accuracy gain on Reward-Bench. Experiments also demonstrate the policy induced by RRM outperforms RM and ODIN in MT-Bench and AlpacaEval-2 benchmark. Analysis show that length artifact and deliberately designed artifact can be eliminated.

**Strengths:**

Reward hacking is an important problem in LLM alignment. This paper introduces causal inference framework in reward modeling to mitigate reward hacking, which offers an interesting perspective. This idea makes a lot of sense since the essence of reward hacking mitigation is to find out the irrelevant artifacts that have no causal relationship with the preference label, and causal inference algorithms just offer a useful tool. The idea is clearly presented and illustrated. The experiment results verify the effectiveness of proposed method.

**Weaknesses:**

The experiments are insufficient. First, the experiments in this paper only cover two possible artifacts, namely length and a deliberately designed phrase, which is not compatible with author's claim that this method should be capable of handling "all potential exploitation patterns". More patterns such as format and emojis should be investigated in the experiments. Second, the reward model is trained only using Gemma-2-9b-it. More experiments should be conducted on different model types such as llama, and larger model sizes such as 57B and 72B. It is in doubt that when the model size increases, the reward hacking problem can be naturally mitigated and the effectiveness brought by this method may diminish. Third, in terms of policy evaluation, this method is only verified in DPO training whereas various other methods should also be considered such as PPO, whose performance relies heavily on reward model quality. Last, the author is suggested to add more baselines in experiment, such as reward model ensemble which can also alleviate reward hacking.

The data augmentation method is trivial. The new combination is derived by randomly selecting answers from other prompts. Since this negative sample is not related to the prompt with high probability, it is too easy for the model to distinguish the positive and negative answers. The author is suggested to incorporate some hard negatives to make the training more robust.

**Questions:**

1. Explain why it is needed to add neutrals and set both non-contextual responses as tie in Section 3.2. What is the necessity of this step?
2. In Table 2, why the performance of RRM is not consistent in two chat datasets such as Chat and Chat Hard? In Chat dataset, RRM outperforms RM while in Chat Hard dataset, RRM is even worse.

---

> ### Author Response · Authors · 2024-11-21
>
> We thank the reviewers for their constructive feedback and valuable insights. Below, we address each of the main concerns raised.
>
> > Re: First, the experiments in this paper only cover two possible artifacts, namely length and a deliberately designed phrase, which is not compatible with author's claim that this method should be capable of handling "all potential exploitation patterns". More patterns such as format and emojis should be investigated in the experiments.
>
> We acknowledge that the artifacts covered in this paper are limited. To further investigate the effectiveness of our approach, we conduct a follow-up analysis as follows:
> With p=0.1, add bold face (wrap the whole response as bold face) to the chosen responses.
> After step 1, with p=0.1, further add emoji to the end of the chosen responses.
> Train RM and RRM and do the same analysis as Figure 5.
> We add a section “Additional Analysis with Mixed Artifacts” in Appendix including the above analysis.
>
> > Re: Second, the reward model is trained only using Gemma-2-9b-it. More experiments should be conducted on different model types such as llama, and larger model sizes such as 57B and 72B. It is in doubt that when the model size increases, the reward hacking problem can be naturally mitigated and the effectiveness brought by this method may diminish.
>
> We didn’t run Llama 3 8B due to policy reasons (https://huggingface.co/meta-llama/Llama-3.1-8B/blob/main/LICENSE). To verify the effectiveness of our approach on a different model size, we conduct experiments on Gemma-2-2b-it (Added to “Additional Results with Gemma-2-2b-it” in Appendix). For larger models such as 57B and 72B, we lack computation resources to conduct the experiment at that scale. Regarding the large models, the GPT-4 judge (a large sized model) is found to have length bias when rating sxs responses [1].  It indeed suggests that the bias pattern does not diminish as model size increases.
>
> [1] Zheng, Lianmin, et al. "Judging llm-as-a-judge with mt-bench and chatbot arena." Advances in Neural Information Processing Systems 36 (2023): 46595-46623.
>
> > Re: Third, in terms of policy evaluation, this method is only verified in DPO training whereas various other methods should also be considered such as PPO, whose performance relies heavily on reward model quality. Last, the author is suggested to add more baselines in experiment, such as reward model ensemble which can also alleviate reward hacking.
>
> Besides DPO, we also evaluate the policy of BoN. [1] shows that BoN policy performs similarly and is more KL efficient than PPO. BoN relies heavily on reward model quality and thus we believe that our evaluation on aligned policy is convincing.
> Regarding reward model ensemble, we argue that it is orthogonal to our work. We focus on debiasing the artifacts learned in the reward model, it is more on mitigating the bias. But the reward model ensemble is more on reducing the variance. It can not eliminate the artifact bias (such as length, emoji, format) existing in the reward model. Besides, our approach can be naturally combined with the reward model ensemble approach.
>
> [1] Gao, Leo, John Schulman, and Jacob Hilton. "Scaling laws for reward model overoptimization." International Conference on Machine Learning. PMLR, 2023.

---

> > ### Author Response · Authors · 2024-11-21
> >
> > > Re: The data augmentation method is trivial. The new combination is derived by randomly selecting answers from other prompts. Since this negative sample is not related to the prompt with high probability, it is too easy for the model to distinguish the positive and negative answers. The author is suggested to incorporate some hard negatives to make the training more robust.
> >
> > The data augmentation indeed appears to be trivial, but we argue that the approach is backed by a solid and sophisticated causal framework. To our knowledge, no previous work has been proposed with this simple approach. Instead, we argue that the “triviality” of our approach is indeed the advantage instead of a disadvantage as it is shown to be quite effective on extensive numerical experiments and analysis.
> > We fully agree with the reviewer that “it is too easy for the model to distinguish the positive and negative answers.”, that’s what the experiment section (line 337-342) has also covered. We construct the “hard negatives” by applying the filter such that the original RM performs badly. We appreciate the reviewer mentioning the “hard negatives”. In the Conclusion section (line 539), we also mentioned that “Future work will explore filtering augmented pairs and matching artifacts when constructing response pairs, further refining the training process.” Thus we believe that a better way to mine the hard negatives can be quite valuable. But we believe that this work opens a door on debiasing RM with a data augmentation approach with a causal learning framework. The hard negative mining can be a natural follow-up of this paper. To further address this, we added “Discussion on Data Filtering Strategies” in Appendix in the updated draft.
> >
> > > Re: Explain why it is needed to add neutrals and set both non-contextual responses as tie in Section 3.2. What is the necessity of this step?
> >
> > As described in Section 3.2 (line 221-227), we consider all 45 possible augmented triplets. The labeling rule is described in line 234-237. We don’t have a prior that neutrals will work but we believe the labeling rule is fair since both responses are off-topic and thus the preference is independent of contextual-signal S. As a result, the winning probability should be 0.5 according to the causal rule we proposed. As shown in Table 3, Neutrals can contribute to the improvement of AlpacaEval-2 and MT-Bench first turn. In reality, users can try with and without neutrals to select a better one with minimal cost since the neutral data can be easily filtered by checking the preference label.
> >
> > > Re: In Table 2, why the performance of RRM is not consistent in two chat datasets such as Chat and Chat Hard? In Chat dataset, RRM outperforms RM while in Chat Hard dataset, RRM is even worse.
> >
> > The Chat is a relatively easy task compared to the Chat Hard. For instance, on MT-bench, the authors of RM-BENCH use 10s v.s. 1s to construct the comparison pairs for the Chat category, while use 7-8 v.s. 5-6 to construct the Chat Hard category.
> > Consequently, the margin between the two responses in the Chat category is large and whether we mitigate the style bias or not will not significantly influence the reward model performance. Therefore, both the RM and RRM perform very well on the Chat task, where the difference may result from the training randomness (e.g. the random initialization). The performance is nearly saturated.
> > In contrast, for Chat Hard, as described in Section 4.1 in [1], “Hard Accuracy on RM-BENCH is significantly lower than Normal Accuracy, with most reward models failing to exceed random-level performance (50%). This reveals that many existing reward models are more akin to style preference models, favoring well-structured responses over those with stronger substantive content. Our findings highlight the urgent need to mitigate style bias and improve the robustness of reward models.” That’s why our approach can perform significantly better. This is strong evidence that we successfully mitigated style bias with a more robust reward model. RRM has 15% higher accuracy than the baseline RM on the Chat Hard dataset.
> >
> > [1] Liu, Yantao, et al. "RM-Bench: Benchmarking Reward Models of Language Models with Subtlety and Style." arXiv preprint arXiv:2410.16184 (2024).

---

> > ### Comment · Reviewer_1CBu · 2024-11-28
> >
> > Thanks for the response from the authors. Some of concerns are addressed in the response, but I still doubt the generalization of the proposed method where the experiments is this paper not cover. I decide to raise my rating to 5.

---

### Official Review · Reviewer_jPJZ · 2024-11-02

**Soundness:** 4
**Presentation:** 4
**Contribution:** 4
**Rating:** 8
**Confidence:** 4

**Summary:**

This paper presents a method to enhance the robustness of the reward model for LLM alignment. More specifically, rewards models are trained based on the question and the human labelers' preference over the answer pairs. A commonly observed un-robust behavior of the reward model is that there are spurious correlations between the label and the artifacts in the answers which is not relevant to the question. The existence of such spurious correlations causes the reward model to favor certain types of artifacts, e.g. the length of the answer.

In previous works, some of the artifacts are specifically dealt with, like introducing a penalty for the length of the answer. However, the solutions lack a principle and can not handle other artifacts like the word preference and style of the answers. In this work, a systematic approach is proposed based on a causal framework, which is effective in debiasing the reward model from various artifacts.

**Strengths:**

- The causal framework has been effective and widely used in other problems, but this is the first time I see it applied in LLM reward modeling, and it very well suits the problem under consideration.
- The implementation is very straightforward, just augment the data, no empirical tricks.

**Weaknesses:**

- It seems suboptimal to me to train the reward model that is only dependent on the S, because the A part could contribute nontrivially to the reward function, in the case where both answers are equal in terms of the contextual quality, the labeler could prefer the answer which is organized as bullet points for readability.
- The explanation on why "reasoning" is performing worse under RRM is not quite convincing. Even if they are less affected by non-contextual artifacts, the augmented data should not have caused the degrade of performance either (as the augmented data are simpler pairs). I would suggest two other hypothesis to investigate:
  - Look into how "-Neutral" works on "Reasoning" as requiring 50/50 preference over some random pairs could be too strong.
  - Try to add back the "A only" reward model, as mentioned above the artifact could contribute nontrivially to the reward, removing it entirely could be detrimental especially for math and code, where a high quality answer should be good in formatting which is noncontextual.

**Questions:**

- The same augmentation and disentangle technique can be applied to learn a artifact only model. Some of the artifacts would improve the quality of the answer, if there is a consensus preference on such artifacts in the labelers. Is there any proposal to combine the contextual & non-contextual models? Would adding these two reward models work, or do we need to take the min of these two models? Any suggestions from the authors?

---

> ### Author Response · Authors · 2024-11-21
>
> We thank the reviewers for their constructive feedback and valuable insights. Below, we address each of the main concerns raised.
>
> > Re: It seems suboptimal to me to train the reward model that is only dependent on the S, because the A part could contribute nontrivially to the reward function, in the case where both answers are equal in terms of the contextual quality, the labeler could prefer the answer which is organized as bullet points for readability.
>
> We agree that responses with better organization, such as those formatted with bullet points, should be preferred in general. But we would like to address that conditioning on prompt, there can be richer information on determining the preference. There can be cases that the prompt specifies “please do not include bullet points in your response”. Even for factuality and safety attributes, the prompt can ask the model to create a virtual horror story which may violate the factuality/safety constraints.
> Regarding “in the case where both answers are equal in terms of the contextual quality, the labeler could prefer the answer which is organized as bullet points for readability.”, we have a few thoughts to share:
>
> 1. We would like to answer the question that “what the preference will be if both responses share the same artifacts” following length-controlled Alpaca [1]? Thus if we modify the one answer to remove the “bullet points”, what the preference will be? Maybe after removing the bullet points, the contextual quality can change, becaues the prompt can indicate whether bullet points are needed. If the prompt is for creative writing, removing bullet points can even improve the preference.
>
> 2. Even if there is such a pattern that should be uniformly preferred, our RRM can still learn this pattern because we only augment data, not delete any existing data. The purpose of adding augmented data is to force the model to learn better and avoid native shortcut prediction.  The second point is, theoretically our proposed method can completely remove the effect of A, when we have unlimited samples. But in practice, we don't have infinite samples, and due to computation budget, we only add a small portion of fully augmented data distribution. The added augmented data helps improve the robustness.
>
> 3. We think the reviewer raises the valid point on prompt-agnostic artifacts that can uniformly contribute to the preference such as safety. To address this, in the conclusion section, we mention that “Future work will explore filtering augmented pairs and matching artifacts when constructing response pairs, further refining the training process.”. Thus, in the future, we should apply filters on the prompt and responses to guarantee that the losing responses are not too bad. But we believe that our work opens a door on debiasing RM with a data augmentation approach with a causal learning framework. Follow-up works can focus on matching the effects of chosen/rejected responses, and filtering data by prompt/responses.
>
> [1] Dubois, Yann, et al. "Length-controlled alpacaeval: A simple way to debias automatic evaluators." arXiv preprint arXiv:2404.04475 (2024).

---

> > ### Author Response · Authors · 2024-11-21
> >
> > > Re: The explanation on why "reasoning" is performing worse under RRM is not quite convincing. Even if they are less affected by non-contextual artifacts, the augmented data should not have caused the degrade of performance either (as the augmented data are simpler pairs). I would suggest two other hypothesis to investigate:
> > > * Look into how "-Neutral" works on "Reasoning" as requiring 50/50 preference over some random pairs could be too strong.
> > > * Try to add back the "A only" reward model, as mentioned above the artifact could contribute nontrivially to the reward, removing it entirely could be detrimental especially for math and code, where a high quality answer should be good in formatting which is noncontextual.
> >
> > Let's address the above one-by-one.
> >
> > > Re: The explanation on why "reasoning" is performing worse under RRM is not quite convincing
> >
> > After a careful examination of the RewardBench dataset, we realize that the reasoning test samples are generated in an asymmetrical way. Specifically, all the chosen responses are generated by humans, while all the rejected responses are generated by GPT-4. Eventually, models can learn the spurious features of the format to cheat in the test. For instance, in the math-prm subset, the average number of characters in the chosen response is 524.6, while it is 1213.1 for the rejected responses. As our method tries to mitigate the related biases, the decline in the reasoning accuracy is also expected. Therefore, the high reasoning accuracy of the RM cannot reflect the real performance advantage. This is a weakness from the evaluation benchmark, and it would be interesting to build a more reliable test set in the future.
> >
> > > Re: Look into how "-Neutral" works on "Reasoning" as requiring 50/50 preference over some random pairs could be too strong.
> >
> > This is a great suggestion. We evaluated the Reasoning on “-Neutral” RRM, and the score decreased from 90.62 to 86.27. We hypothesized that it was because the reasoning prompt set is relatively small and the augmented non-contextual data can be too easy (with one reasoning and one non-reasoning) so that the augmented data are more likely to be filtered by the original RM. With Neutral, at least both responses are reasoning related or the prompt is reasoning related, thus it may balance the proportion of reasoning data in the data mixture. But we agree that assigning 50/50 needs a careful filtering on the relevance of the response to the prompt. Sometimes it can be too strong.
> >
> > > Re: Try to add back the "A only" reward model, as mentioned above the artifact could contribute nontrivially to the reward, removing it entirely could be detrimental especially for math and code, where a high quality answer should be good in formatting which is noncontextual.
> >
> > As explained above, the validation set on reasoning has strong style bias: the chosen answers are written by human experts and rejected answers are generated by LLM, which has a strong style. This indicates “A only” model can be useful. However, our work mainly focuses on chatting prompts for improving instruction-following and quality. Besides, we still include the original data in our training, which suggests that the noncontextual information can still be learned. We argue the “A only” model may better fit as a pointwise reward, and the augmentation works for pairwise reward. But we need careful filtering after the augmentation. We added a section “Discussion on Data Filtering Strategies” in the Appendix further discussing this.
> >
> > > Re: The same augmentation and disentangle technique can be applied to learn a artifact only model. Some of the artifacts would improve the quality of the answer, if there is a consensus preference on such artifacts in the labelers. Is there any proposal to combine the contextual & non-contextual models? Would adding these two reward models work, or do we need to take the min of these two models? Any suggestions from the authors?
> >
> > This is a great question. We suggest that the reward model can include multiple attributes/dimensions. The RRM works especially well for instruction following and helpfulness. For safety or other prompt-independent dimension, we can use another head to predict that as done in ODIN [1].
> >
> > [1] Chen, Lichang, et al. "Odin: Disentangled reward mitigates hacking in rlhf." arXiv preprint arXiv:2402.07319 (2024).

---

### Official Review · Reviewer_Vrzq · 2024-11-03

**Soundness:** 3
**Presentation:** 3
**Contribution:** 3
**Rating:** 8
**Confidence:** 4

**Summary:**

This paper aims to learn reward models that are unbiased by “artifacts” that come from human preference data: namely length biases and formatting. To do so, they propose to augment the RLHF preference dataset with preference pairs where one sample is in response to the prompt, and the other example is not in response to the prompt. They evaluate their learned reward model on reward bench and use it to improve downstream policy performance.

**Strengths:**

- The authors study a very relevant and timely problem.
- The paper shows significant improvement in both reward modeling accuracy and in downstream policy performance.
- The authors do a really interesting study where they artificially add artifacts to the preference dataset, and find their method is more robust to it.

**Weaknesses:**

- The main weakness of this paper is that the experiments are not that comprehensive. Although the results are strong, the experiments are only conducted on one dataset, with one model, and with one random seed. Therefore we cannot tell if the results are statistically significant or generalizable.
- The baseline is pretty weak with regards to reward bench. RLHFFlow with Llama 3 8B gets 87.1 on reward bench, while the author’s baseline gets 80.
- The writing describing the methodology not that clear. I think more plainly describing the data augmentation would be better.
- The data filtering step used in their method hurts their experiments interpretability, as it may be the case that the sole reason RRM gets good performance is due to the data filtering step. They should include a baseline where the original dataset is filtered, and then a RM is trained based on that.

**Questions:**

- How did the authors tune the hyperparameters?
- What are the “three possible prompts” described in line 226?

---

> ### Author Response · Authors · 2024-11-21
>
> We thank the reviewers for their constructive feedback and valuable insights. Below, we address each of the main concerns raised.
>
> > Re: The main weakness of this paper is that the experiments are not that comprehensive. Although the results are strong, the experiments are only conducted on one dataset, with one model, and with one random seed. Therefore we cannot tell if the results are statistically significant or generalizable.
>
> We acknowledge that the experiments are not comprehensive enough, although we have evaluated thoroughly on RewardBench, MTBench, and AlpacaEval2 with various alignment algorithms (DPO/BoN) and baselines (ODIN/RM). To further enhance the confidence of our results, we add experiments with a Gemma-2-2b-it reward model in Appendix (Additional Results with Gemma-2-2b-it) of the updated draft. To understand better on whether the artifacts can be effectively mitigated, we also add experiments with two additional artifacts: bold faces and emojis in Appendix (Additional Analysis with Mixed Artifacts). With additional experiments, it enhances the significance of our framework.
>
> > Re: The baseline is pretty weak with regards to reward bench. RLHFFlow with Llama 3 8B gets 87.1 on reward bench, while the author’s baseline gets 80.
>
> We didn’t run Llama 3 8B due to policy reasons (https://huggingface.co/meta-llama/Llama-3.1-8B/blob/main/LICENSE). We use the same dataset as those in RLHFFlow and for baseline and our approach, we use the same model (Gemma-2-9b-it) as a fair comparison. To further verify the results, we add a Gemma-2-2b-it in Appendix (Additional Results with Gemma-2-2b-it).
>
> > Re: The writing describing the methodology not that clear. I think more plainly describing the data augmentation would be better.
>
> We thank the reviewer for the feedback. We illustrate our augmentation strategy in Figure 1 for illustration purpose. We describe our approach in plain language in line 086 as “Our pipeline is illustrated in Figure 1, where we augment the reward model training data by using responses from other examples to effectively balance the artifacts in chosen and rejected responses”. In line 224, we also mention that “In practice, we can shuffle the dataset twice to achieve $σ_1$ and $σ_2$.”. We can further improve the clarity of our approach in the camera-ready version.
>
> > Re: The data filtering step used in their method hurts their experiments interpretability, as it may be the case that the sole reason RRM gets good performance is due to the data filtering step. They should include a baseline where the original dataset is filtered, and then a RM is trained based on that.
>
> Sorry for the confusion. In the paper we mentioned that “To reduce the augmented data size, we first conduct inference on random 50% of the augmented data using the trained RM, …”. We only filter the data on the augmented triplets and do not filter the original data. So in both baseline and our approach, we use the full copy of the original data.
>
> > Re: How did the authors tune the hyperparameters?
>
> In line 342-344 (and footnote 14) and line 350-352, we cover how to tune the hyperparameters. We use grid search to pick the best hyperparameter based on evaluation metrics (evaluation accuracy for RM and AlpacaEval2 for RLHF).
>
> > Re: What are the “three possible prompts” described in line 226?
>
> The three possible prompts correspond to $(x^{(i)}, x^{(\sigma_1(i))}, x^{(\sigma_2(i))})$, which corresponds to the original prompt, a prompt from shuffled dataset, and a prompt from a twice-shuffled dataset.

---

> > ### Comment · Reviewer_Vrzq · 2024-11-21
> > **Thank you for the rebuttal**
> >
> > Thank you for the rebuttal, it improved my understanding of the paper. All of my concerns are addressed.
> >
> > I still suggest the authors improve upon the writing (particularly section 3.2). I think that the "Possible Combinations" paragraph (line 221) could use some simplification. After reading the paper a few times it is clear what is meant, but when someone is first reading the paper it seems like an overcomplicated way to explain a simple idea.

---

> > > ### Author Response · Authors · 2024-11-21
> > >
> > > Thank you for your prompt response and thanks for the suggestion! We will improve the writing in our camera-ready version.

---

### Official Review · Reviewer_gp5Y · 2024-11-04

**Soundness:** 3
**Presentation:** 3
**Contribution:** 2
**Rating:** 5
**Confidence:** 5

**Summary:**

Traditional reward model (RM) training for large language models (LLMs) struggles to separate prompt-driven preferences from irrelevant artifacts like response length and format. This work introduces a causal framework and data augmentation technique to remove these artifacts, resulting in a robust reward model (RRM) that improves performance across benchmarks, enhancing accuracy and alignment scores significantly.

**Strengths:**

This paper addresses an important issue in preference datasets: how to disentangle prompt-driven preferences from prompt-independent artifacts.

**Weaknesses:**

While the problem studied in this paper is significant, the proposed solution is relatively straightforward, focusing only on expanding the dataset. To remove prompt-independent artifacts, the dataset size needs to be several times larger, increasing the training cost. Additionally, as shown in Figure 5, the effect of this costly augmentation is not very significant—after inserting 30% artifacts, the RRM only decreases from the original RM’s 25% to 20%.

Regarding Table 1, the reasoning performance declines, and the authors explain this as due to math and coding tasks being less affected by non-contextual artifacts. This raises the question of whether the proposed method might have a negative impact on attributes unaffected by non-contextual artifacts.

**Questions:**

The method proposed in this paper to 'disentangle prompt-driven preferences from prompt-independent artifacts' seems to only eliminate biases unrelated to the input while preserving input-related preferences. This can effectively reduce bias under the 'helpful' alignment objective. However, if our alignment goal is 'safety', the response might be prompt-independent, typically just refusing to reply. Yet humans may still have preferences for different ways of refusal. In this case, how can we disentangle prompt-independent preferences from prompt-independent artifacts?

---

> ### Author Response · Authors · 2024-11-21
>
> We thank the reviewers for their constructive feedback and valuable insights. Below, we address each of the main concerns raised.
>
> > Re: While the problem studied in this paper is significant, the proposed solution is relatively straightforward, focusing only on expanding the dataset. To remove prompt-independent artifacts, the dataset size needs to be several times larger, increasing the training cost. Additionally, as shown in Figure 5, the effect of this costly augmentation is not very significant—after inserting 30% artifacts, the RRM only decreases from the original RM’s 25% to 20%.
>
> > Re: While the problem studied in this paper is significant, the proposed solution is relatively straightforward, focusing only on expanding the dataset. To remove prompt-independent artifacts, the dataset size needs to be several times larger, increasing the training cost.
>
> We acknowledge that our proposed solution may seem straightforward, as it involves dataset expansion. However, this simplicity belies the sophistication of the underlying causal learning framework, a novel theoretical approach not previously applied to this challenge. By disentangling prompt-independent artifacts from input-driven preferences, our method achieves both simplicity and effectiveness. This enables a broader range of applications across diverse use cases.
> Regarding training cost, we recognize the concern but argue that the cost increase remains manageable within the broader context of reinforcement learning with human feedback (RLHF). The reward model (RM), while crucial to alignment, incurs comparatively less computational burden than supervised fine-tuning or policy optimization. Thus, improving the RM, the key component for model alignment, justifies the additional investment in training. In our study, we trained both the RM and the robust RM (RRM) to convergence, fully utilizing the value and information embedded in our expanded dataset to ensure optimal performance.
>
> > Re: Additionally, as shown in Figure 5, the effect of this costly augmentation is not very significant—after inserting 30% artifacts, the RRM only decreases from the original RM’s 25% to 20%.
>
> We understand that the observed improvement may initially appear modest. However, it’s essential to consider the context of the evaluation. With 30% artifacts introduced into responses, a random baseline would yield around 30% artifact presence in optimal responses. Thus, achieving 25% with RM and 20% with RRM represents a meaningful improvement compared to the baseline. The reduction of 5 percentage points below this baseline signifies effective artifact mitigation, illustrating the method’s impact in refining alignment robustness.

---

> > ### Author Response · Authors · 2024-11-21
> >
> > > Re: Regarding Table 1, the reasoning performance declines, and the authors explain this as due to math and coding tasks being less affected by non-contextual artifacts. This raises the question of whether the proposed method might have a negative impact on attributes unaffected by non-contextual artifacts.
> >
> > > Re: the reasoning performance declines
> >
> > After a careful examination of the rewardbench dataset, we realize that the reasoning test samples are generated in an asymmetrical way. Specifically, all the chosen responses are generated by humans, while all the rejected responses are generated by GPT-4. Eventually, models can learn the spurious features of the format to cheat in the test. For instance, in the math-prm subset, the average number of characters in the chosen response is 524.6, while it is 1213.1 for the rejected responses. As our method tries to mitigate the related biases, the decline in the reasoning accuracy is also expected. Therefore, the high reasoning accuracy of the RM cannot reflect the real performance advantage. This is a weakness from the evaluation benchmark, and it would be interesting to build a more reliable test set in the future.
> >
> > > Re: the proposed method might have a negative impact on attributes unaffected by non-contextual artifacts.
> >
> > We totally agree with the reviewer that certain non-contextual artifacts may be desired. In our setting, most of the RM datasets are chat, so we care more about instruction following and relevance. Regarding attributes such as code execution, math reasoning correctness, and safety, we agree that this should not be added to the augmentation. There is a trade-off between helpfulness (instruction following) and those attributes. If the rejected responses are not extremely bad so that they can still be helpful in addressing the prompts. A careful filtering on the extreme bad responses is needed. In the conclusion section, we mention that “Future work will explore filtering augmented pairs and matching artifacts when constructing response pairs, further refining the training process.”. Thus, in the future, we should apply filters on the prompt and responses to guarantee that the losing responses are not too bad. Nevertheless, we believe that our work opens a door on debiasing RM with a data augmentation approach with a causal learning framework. Follow-up works can focus on matching the effects of chosen/rejected responses, and filtering data by prompt/responses. To further address this part, we added a section in Appendix (Discussion on data filtering strategies) in the updated version.

---

> > > ### Author Response · Authors · 2024-11-21
> > >
> > > > Re: The method proposed in this paper to 'disentangle prompt-driven preferences from prompt-independent artifacts' seems to only eliminate biases unrelated to the input while preserving input-related preferences. This can effectively reduce bias under the 'helpful' alignment objective. However, if our alignment goal is 'safety', the response might be prompt-independent, typically just refusing to reply. Yet humans may still have preferences for different ways of refusal. In this case, how can we disentangle prompt-independent preferences from prompt-independent artifacts?
> > >
> > > We thank the reviewer for highlighting this important consideration. We fully acknowledge this limitation. In the context of this work, our primary focus was on chat-oriented prompts where responses generated by large language models were not extremely unsafe, and thus, this approach was effective in our evaluation. In real applications, we agree that safety alignment requires careful handling of prompt-independent preferences. In future iterations, we will refine our augmentation process to include additional safety layers, ensuring that rejected responses do not introduce undue risk. To further address this part, we added a section in Appendix (Discussion on data filtering strategies) in the updated version.

---

### Meta-Review · Area_Chair_wiex · 2024-12-24

**Metareview:**

This paper proposes a robust reward model (RRM) training method to mitigate reward hacking by leveraging a causal inference framework. A causal graph for human preference modeling is introduced to help the model distinguish between contextual preference signals and context-free artifacts. Guided by the causal inference framework, the training data is augmented by reorganizing the (prompt, positive, negative) triplet. Experiments demonstrate that the proposed approach effectively filters out undesirable artifacts during reward model training, resulting in a more robust reward model. Specifically, when training the reward model on Gemma-2-9b-it, RRM achieves an absolute 3.54% accuracy improvement over RM on Reward-Bench. Furthermore, policies induced by RRM outperform those trained with RM and ODIN on the MT-Bench and AlpacaEval-2 benchmarks. Analysis also reveals that artifacts such as length bias and deliberately designed artifacts can be effectively eliminated.

The strengths of this work lie in:
- Its focus on addressing a critical issue in reward model training.
- The novel application of a causal inference framework to tackle this issue.

Given these two strengths, I recommend acceptance at the current stage, albeit with moderate confidence, due to the following reasons:
- Almost all reviewers raised concerns about the generalization of the proposed method to more capable LLMs, as experiments on LLMs other than Gemma or larger-scale models are missing. While the authors added results for Gemma-2-2b-it in the Appendix, they are strongly encouraged to include additional experiments on larger-scale LLMs before the camera-ready submission.
- Both Reviewer gp5Y and Reviewer jPJZ expressed concerns about the reasoning performance drop observed with RRM. Although the authors explained this is due to the test set, it is strongly recommended that they include results from another test set to further evaluate RRM's reasoning performance before the camera-ready version.

Providing more comprehensive empirical results rather than limiting the work to a proof-of-concept would significantly broaden the scope and appeal to a wider audience.

**Additional Comments On Reviewer Discussion:**

During the discussion period, Reviewer Vrzq and Reviewer 1CBu actively engaged with the authors. Reviewer Vrzq expressed satisfaction with the responses provided, while Reviewer 1CBu's remaining concern primarily focused on the generalizability of the proposed method to LLMs beyond Gemma and to larger-scale models.

The strengths and weaknesses of this work have been summarized above, resulting in my current recommendation for acceptance, albeit with moderate confidence.

---

### Decision · Program_Chairs · 2025-01-22

Accept (Poster)